# Parapharyngeal Space Tumors: Our Experience

**DOI:** 10.3390/jpm13020283

**Published:** 2023-02-02

**Authors:** Jacopo Galli, Rolando Rolesi, Roberto Gallus, Annalisa Seccia, Alessandro Pedicelli, Francesco Bussu, Emanuele Scarano

**Affiliations:** 1Fondazione Policlinico Universitario A. Gemelli IRCCS, 00168 Rome, Italy; 2Institute of Otolaryngology, Università Cattolica Sacro Cuore, 00168 Rome, Italy; 3Division of Otolaryngology, Mater Olbia Hospital, 07026 Olbia, Italy; 4ENT Division, Department of Medicine, Surgery and Pharmacy, University of Sassari, 07100 Sassari, Italy; 5Division of Otolaryngology, Azienda Ospedaliera Pia Fondazione di Culto e Religione Cardinale G. Panico, 73039 Tricase, Italy

**Keywords:** para-pharyngeal space, transoral surgery, SQUID12, Contour, EVOH

## Abstract

Para-pharyngeal space (PPS) tumors include an heterogeneous group of neoplasms, accounting for approximatively 0.5–1.5% of all head and neck tumors. Management of these neoplasms requires a careful diagnostic workout and an appropriate surgical approach to obtain good outcomes associated with minimal aesthetic drawbacks. In this study we investigated clinical onset, histologic features, surgical treatment outcomes, peri operative complications and follow up of 98 patients treated for PPS tumors in our Centre between 2002 and 2021. Furthermore, we reviewed our preliminary experience of preoperative embolization of hyper vascular PPS tumors trough SQUID12, an ethylene vinyl alcohol copolymers (EVOH) which exhibits many advantages over other embolic agents, due to its better devascularization rate and lower risk of systemic complications. Our data support the hypothesis that transoral surgery scenario should be significantly revised, as it could represent a valid treatment for tumors located in lower and prestyloyd portion of PPS. Moreover, SQUID12, a novel embolization agent, may be a very promising choice for PPS hyper vascularized tumors, ensuring higher devascularization rate, safer procedures and lower risk of systemic dispersion compared to traditional Contour treatment.

## 1. Introduction

Tumors arising from para-pharyngeal space (PPS) include a wide and heterogeneous group of lesions which are relatively uncommon, accounting for approximatively 0.5–1.5% of all head and neck neoplasms [1,2,3]. Surgical excision is the primary treatment for these tumors, mainly consisting of trans/mandibular, trans-cervical, trans/parotid, and trans-oral approaches or any combination of these [4]. There is not a unique optimal treatment for all PPS tumors, due to several reasons. First of all, the variable tumor location inside PPS, which is a very difficult space to be reached due to an high anatomical complexity [5,6] containing many vital structures such as carotid artery, cranial nerves or jugular veins. Moreover, the possible high tumor size and its biological characteristics (e.g., histopathology, vascularization and relationship with critical PPS neuro vascular structures) must be taken into account [7,8]. An appropriate surgical treatment should achieve an optimal anatomic exposure and complete tumor resection as well as minimal peri-operative complications and postoperative aesthetic/functional morbidity. For these reasons, the surgical treatment of these lesions and the choice of the most appropriate surgical technique is often a great challenge, especially for tumors located in the superior portion of PPS, due to its proximity to the skull base region. Thus, in the last two decades a growing interest focused on appropriateness of current surgical indication and on the development of minimal invasive surgical techniques [9,10,11,12,13,14].

In this study we investigated clinical onset, histologic features, surgical treatment outcomes, peri operative complications and follow up of 98 patients treated for PPS tumors in our Centre between 2002 and 2021. Furthermore, we reviewed our preliminary experience of preoperative embolization of hyper vascular PPS tumors trough SQUID12, an ethylene vinyl alcohol copolymers (EVOH) which exhibits many advantages over other embolic agents, due to its better devascularization rate and lower risk of systemic complications.

## 2. Materials and Methods

We performed a monocentric, retrospective observational study, on a total of 1800 patients affected by cervical tumors, who underwent to diagnostic and therapeutic workout in our Centre between 2002 and 2021. Patients were identified considering a list of International Classification of Disease (ICD) codes that refer to neoplasms involving PPS. According to many inclusion criteria of Riffat et al., (2014) [1], we selected 98 patients, 50 males and 48 females, ranging from 20 months to 86 years of age. More specifically, only tumors located in PPS were considered eligible, while metastatic neoplasms involving PPS but originating from other anatomical regions were excluded. Among the salivary neoplasms of the parotid deep lobe, we included only those located in the retroangulomandibular neck region. Among the carotid paragangliomas, we considered eligible only those located above the posterior belly of the digastric muscle. Cases lacking an accurate description of preoperative diagnostic techniques, adequate report on the tumor characteristics, surgical approach, histological subtypes diagnosis, postoperative outcome or complications accurate report were excluded. Diagnostic workout revealed bilateral neoplasms in two patients, so a total of 100 PPS tumors, ranging from 16 to 83 mm maximum diameter and involving pre or retro stiloyd parapharingeal spaces, were studied. Compressively, 96 cases underwent surgery. One patient was exclusively treated with embolization and three more cases with an elective radio-chemiotheraphy protocol after trans cervical or endoscopic biopsy to confirm histological diagnosis. surgical planning was performed according to tumor location, histology, dimensions, imaging data, surrounding anatomic features and clinical findings.

### 2.1. Preoperative Evaluations

All patients underwent a fully preoperative evaluation, including general clinical assessment, haematological, cardiovascular and pneumologic evaluations. Although MRI is generally considered the best choice in the evaluation of parapharyngeal tumors as it usually provides better information about soft tissues compared to TC scan, we performed both MRI (76 patients) and CT (51 patients) preoperative examinations in many cases, as they could be complementary in tumor characteristics evaluation, particularly in potentially malignant lesions. MRI was often more able to precisely define the relationship between tumor and surrounding great vessels, especially internal carotid artery (ICA) and to define more accurately the nature of some specific neoplasms (e.g., pleomorphic adenomas, paragangliomas). On the other hand, CT scans provided more accurate information on bony invasion, erosion or calcification within tumors. A selective angiography was preoperatively proposed for all enhancing lesions (37 patients) to favor differential diagnosis between neurogenic and vascular tumors. Ultrasound-assisted fine needle aspiration (USgFNAC) was performed when malignancy was clinically or radiologically suspected (25 patients), with the exception of vascular lesions or in case of inadequate ultrasound targeting or lesion proximity with major vessels. Moreover, in patients who received malignant diagnosis, FDG-PET has been performed as a useful method to detect distant metastasis not only in pre-operative assessment to support treatment planning, but also in mid and late post-operative follow-up. Incisional transoral or transcervical biopsy was rarely considered beneficial in our patients, thus it was performed only in patients who were not candidates for surgery, to obtain the histological diagnosis (3 patients).

### 2.2. Embolization Treatments

Endovascular treatment was performed under general anaesthesia on a total of 16 patients, using Allura Xper FD 20 angiographic system (Philips Medical System, Eindhoven, the Netherlands). In all cases, a catecolamine secretion test and preoperative angiographic study were performed to exclude major contraindications and precisely assess tumour vascularization characteristic (both feeding arteiries and draining veins). Eight patients were treated, between 2010 and 2016 with Polivinil Alcohol (PVA) Contour micro particles (Boston Scientific, Boston MA) while 8 patients, treated between 2016 and 2020, PPS tumors were embolized through SQUID12 direct injection. Puncture procedure was done under ultrasound and fluoroscopic guide, inserting a 19–22 gauge, 10 mm needle directly in the specific tumor target vessels. Needle hub was then connected to DMSO-compatible tube, to allow the injection of embolic liquid agent. A specific balloon micro catheter (SCEPTER 4 × 11 mm XC, Microvention CA, USA) was positioned, to prevent external carotid artery reflux during SQUID12 injection. A post procedure angiography was performed to assess the extent of tumor devascularization, graded as total (100%), near total (95–99%), sub-total (70–95%), moderate (30–70%), low (up to 30%).

### 2.3. Statistical Data

Data analyses of examination, clinical features, diagnoses, surgical approaches, outcomes are presented as means ± standard errors of the mean (SEM) and differences were assessed using ANOVA variance analysis (Statistica, Statsoft, Tulsa, OK, USA); *p*-value < 0.05 was considered significant.

## 3. Results

### 3.1. Clinical Features

In most cases (38%), main clinical onset was aspecific and characterized by foreign body sensation in pharynx. In 17% of cases, patients were completely asymptomatic. Another common symptom was pain (23%) followed by facial nerve weakness (7%), X nerve weakness (4%), dysphagia (8%), nasal obstruction (7%), dysphonia (5%), headache (3%). Other less frequent symptoms were otalgia, tinnitus, dizziness and fullness. (See Figure 1A for more details).

### 3.2. Hystopathologic Diagnoses

Among our sample of 100 cancers, 97 were primary whereas 3 were secondary tumors (specifically, 1 larynx carcinoma metastasis, 1 kidney carcinoma metastasis, 1 gastric cancer metastasis). As for histological features, we found a total 69 benign and 28 malignant ones (71% vs 29%), after excluding metastases. The most frequent benign tumors were pleomorphic adenoma (n = 28), paraganglioma (n = 20), Schwannoma (n = 11) while the most frequent malignant tumors resulted primary squamous cell carcinoma of the parotid gland (n = 10), carcinomas within pleomorphic adenoma (n = 6), myoepithelial carcinomas (n = 5). (See Figure 1B for more details).

### 3.3. Surgery

Tumor resection was done in all patients underwent surgery. surgical approach was individually planned according to tumor size, location, histology diagnosis, and relationship with other PPS anatomical structures. The maximum medium diameter of tumors was 51.8 mm. Transcervical approach was the most frequent choice in our sample (55 cases, 57%), followed by transoral (26 cases, 27%) and trans mandibular surgery (15 cases, 16%). More specifically, transcervical approach was preferred to treat 55 patients affected by benign PPS tumors, located in the lateral and lower portion of PPS space.

Transoral approach was performed in 26 patients to treat 23 benign PPS tumors (17 pleomorphic adenomas, 2 Schwannomas, 2 neurofibromas 1 angiomatoid and 1 bronchial cyst) and 3 malignancies (2 squamous carcinoma, 1 myoepithelial carcinoma). Tumor maximum diameter spared from 18 to 83 mm. All tumors were completely located in inferior and prestiloyd portion PPS. Surgery duration ranged from 31 to 94 min (mean 54.5 min). No significant association between surgery duration and tumor type or dimensions was found. Lastly, transmandibular approach was preferred in patients affected by tumors primarly located or extended in the upper part of PPS space, deep parotid lobe tumors, vascular tumors extended to skull base, malignant tumors. (Table 1).

### 3.4. Post-Operative Complications

Overall, post operative complications occurred in 26 patients. 14 cases were complicated by neurological deficits, while in 12 patients were observed non neurological complications. More specifically, regarding neurological complications among 15 patients who underwent trans mandibular surgery, specific nerve deficits were detected in 5 cases (33%). They showed a significant early-onset post operative disphagia due to homolateral vagal or concomitant hyoglossus and vagal nerve (4 and 1 cases respectively) injury. About transcervical approach, neurological complication rate was significantly lower, interesting 9 out of 55 cases (16%). Facial nerve palsy was the most frequent injury (5 cases underwent deep parotidectomy), followed by spinal nerve (2 cases) and recurrent nerve impairment (2 cases). Conversely, no neurological complications were found in 26 patients treated with trans oral surgery. Non-neurological complications were found in a total of 12 patients. Among cases treated with trans mandibular surgery, most frequent complications were ab-ingestis pneumonia (2 cases), and uncontrolled pain (2 cases). Another patient had a postoperative 6-h persistent confusional state, not related to encephalitis radiological changes or other neurological complications.

In transcervical surgery patients, no extra-neurologic complications were found in early and late post operative period. Among patients treated with trans oral approach, we reported 2 cases of oral bleeding and one case of moderate dyspnea due to upper airways post operative inflammatory status.

Overall, both neurologic and extra-neurologic post operative complications were more frequent in trans mandibular surgery (33 and 27% respectively) as compared to transcervical approach (2 and 16%) or transoral surgery (0 and 12% respectively). The average length of hospitalization was significantly higher in transmandibular surgery with respect to trans-cervical or trans-oral surgery (25.3 days vs 9.9 and 9.3 days respectively, *p* < 0.005). In patients underwent transoral surgery, autonomous alimentation was restored after about 3.7 days, significantly earlier as compared to transcervical (4.8 days, *p* < 0.005) and transmandibular surgery (19.6 days, *p* < 0.001). (See Table 2 for details).

### 3.5. Embolization Procedures Outcomes

Eight patients affected by hyper vascular tumors (6 carotid paragangliomas, 1 vagal paraganglioma, 1 Schwannoma) were preoperatively treated, between 2010 and 2016, with conventional Contour agent, while a total of 8 cases (5 carotid paragangliomas, 2 vagal paragangliomas, 1 Castelman’s follicular lymphoma) treated between 2016 and 2020, were preoperatively embolized whith SQUID12. The mean volume of the PPS tumors was 352 mm^3^ (according to TC and MRI findings,). Mean SQUID12 injection time was 54 min (range 37 to 82 min, evaluated considering time between first and last injection). A mean SQUID12 volume of 22.3 mL was used on each treatment. Devascularization rate was significantly higher in patients treated with SQUID12, reaching the total (100%) or near-total (95–99%) grade in 5 and 3 patients respectively. In contrast, contour devascularization grade was near total in 1 case only, and moderate (30–70%) in the remaining 7 cases. After preoperative embolization all patients underwent surgery. Transmandibular approach was performed in 2 patients preoperatively treated with contour and in 3 patients pre-treated with SQUID12, while 11 patients underwent transcervical surgery (6 patients pre-treated with Contour and 5 patients with SQUID12). Transmandibular Surgery duration was significantly lower in SQUID12 group as compared whith Contour treated patients (218 min vs. 264 min, *p* < 0.005).

## 4. Discussion

Parapharyngeal space tumors are rare, accounting for approximatively 0,5/1,5% of all head and neck neoplasms [1,2,3,4,15,16,17]. The diagnosis and management of these tumors are often a big challenge [18,19,20,21,22,23,24] due to their deep location in a complex PPS anatomy (Figure 2), proximity with nervous and vascular structures, histological subtypes [6,7,8,9].

In our Serie of the 50 men and 48 female who underwent surgery, 71% had benign neoplasms and 29% malignant tumors. Most common symptoms characterizing clinical onset were an aspecific foreign body sensation in pharynx (38%) followed by local pain (23%) and dysphagia (8%). Notably, 17% of patients were completely asymptomatic, confirming that PPS tumors are often difficult to diagnose without an appropriate radiologic workout [18,19,20,21,22,23,24]. Mean tumor size at diagnosis was 50.8 mm.

According to the principal data of the literature [1,3,16,17], we found that 69 out of 97 neoplasms were benign lesions. Among the most frequent histotypes, pleomorphic adenomas accounted 28 cases out of 69 benign tumors, followed by paragangliomas (n = 20) and Schwannomas (n = 11) while the most frequent malignancies were primary squamous carcinoma of parotid gland (10 cases out of 28 malignant neoplasms), carcinoma in ex pleomorphic adenoma (n = 6) and myoepithelial carcinomas (n = 5). (Figure 1B for more details). The main surgery approaches to excise these tumors include a trans-cervical, trans mandibular, trans parotid and transoral approach or a combination of these [23,24,25,26,27,28,29,30,31,32]. All approaches present specific advantages and limitations and their indication must be carefully evaluated, taking into account tumor size, location and superior extent towards skull base, vascularity, as well as patient characteristics [4]. Overall, all approaches should achieve complete tumor excision ensuring vital structure preservation, low rate of postoperative complications and acceptable cosmetic results. In this perspective, it is widely accepted that transmandibular approach should be considered only in case of malignancies or large tumors involving ICA and/or skull base, cervical nervous and vascular structures as well as in any case of large tumors requiring a wide surgical exposure. In fact, this surgery offers an optimal PPS access and possibility to control intraoperative complication. Transcervical approach, the most used surgery for PPS tumors treatment, is suitable for both benign and malignant tumors located in the median and lower portion of PPS as well as benign tumors with a limited extent towards skull base [4]. This approach usually permits a satisfactory surgical exposure with adequate visualization of cranial nerves and cervical great vessels. Combined cervical-parotid approach should be preferred to treat tumors involving parotid deep lobe and/or facial nerve. It is also proposed for retro styloid neoplasms in the middle or upper PPS portion. Transoral approach is probably the most controversial. In fact, it has many advantages such as lower risk of nervous injury (particularly of the facial and inferior alveolar nerves) or development of salivary fistulae. In addition, transoral approach does not produce the major complications, cosmetic injuries or longer hospitalization usually associated with other surgeries [4,29,30,31,32]. On the other hand, transoral approach is often very difficult due to its narrow access with limited tumor visualization and subsequent increased risk of neurovascular damage, bleeding and tumor spillage. For this reason, many authors have limited the use of this approach in the last few decades.

However, our data seem to indicate that this technique, when proposed for the treatment of benign and well capsulated tumors, located in the prestyloid region with limited cranial and superomedial extension, is associated with low rate of peri/operative complications, short hospitalization and good functional and aesthetic outcomes. In fact, in our case series, a complete tumor excision was achieved in all cases treated, without relevant intraoperative bleeding, transient or permanent neurological impairment or other major complications and only in two cases (7.7%) an early surgical revision, during the first postoperative day, was necessary due to moderate local bleeding. The technique has also proved to be particularly rapid, with a mean operation time of 54.5 min and an average hospitalization of 9.3 days, both values significantly lower as compared to other procedures.

In addition, long term recurrence rate was quite low. In fact, during a 18-months/5-years clinical and radiologic follow up, no recurrences were detected in patients originally treated for neurogenic or mesenchymal neoplasms while, regarding salivary neoplasm, we found only 1 recurrence case on 17 pleomorphic adenomas which underwent transoral surgery. This finding appears particularly interesting in light of the possible phenomenon of “tumor spillage” (fragmentation of the neoformation during the resection procedure) which in our series occurred in 65% of cases. These results seem to indicate that, as regards particularly benign tumors located in the prestyloid and lower PPS portion, the recurrence rate is not heavily conditioned by intraoperative tumor fragmentation, but rather by the possibility that the surgical technique adopted allows an adequate completion of the tumor resection, even when en-block removal was not possible.

Another interesting point regards the comparison with the main results obtained in other Centers by transoral robotic surgery (TORS), a relatively new procedure in the treatment of parapharyngiomas. Its indications are substantially similar to those mentioned above for the transoral approach [33,34,35,36,37,38]. The procedure is based on the use of the Da Vinci Surgical Robotic System, a robotic device that combines great operational precision with easy use and handling. For the analysis of the outcomes related to the TORS treatment we took into consideration two reviews which, for endpoints, variables evaluated and sample size, were adequately comparable with our case series [39,40].

Certainly, TORS offers the advantage of shorter post-operative hospitalization and a shorter time to resume autonomous feeding, however the frequency of post-operative complications appears substantially similar, if not even greater than conventional transoral surgery. On the contrary, robotic surgery involves longer operating times (on average 68.8 min vs 54.5 in our series), a greater probability of fragmentation of the tumor mass and a more frequent need to complete the surgical excision by resorting, intra-operatively or subsequently, to transcervical access (15–20% of cases), required in only one case to complete our conventional procedure.

Since these are highly engineered innovative technologies, we believe that it is appropriate to take also into consideration aspects of a managerial-logistic-financial nature. Many analyses focused on the evaluation of the cost-benefit ratio of this surgery, underline that the adoption of the most common robotic surgery systems, including the Da Vinci Surgical System itself, requires high initial economic investment, (approximately 1.5–2 million euros), to be added to high management costs over time, (accounting for approximately 0.1/0.2 million euros/year). This leads many authors [41] to believe that the investment can be justified only within centers with a high annual volume of surgical interventions, which can operate an adequate interdisciplinary instrumentation sharing.

Moreover, our data seem to demonstrate the efficacy of SQUID12 in the preoperative embolic treatment of parapharyngiomas, without significant complications [42]. SQUID12 appears to offer many advantages over the traditional Contour treatment. The first one, administered through a manageable and safe procedure, proved to be significantly more effective than Contour, inducing high grade intra-lesional devascularization, always above 90% and complete in 63% of cases. This result was associated with shorter operative times and lower risks of intraoperative bleeding complications. Furthermore, preoperative embolization through Contour, for which endovascular access is required rather than percutaneous / intralesional, is burdened by a greater theoretical risk of retrograde reflux and systemic dispersion of the drug, occurrences never ascertained in our patients, but described in scientific literature [43,44,45,46].

## 5. Conclusions

Management of PPS neoplasms requires a careful diagnostic workout and an appropriate surgical approach to obtain good outcomes associated with minimal aesthetic drawbacks and low functional impairment. Surgical treatment of PPS tumors is often a great challenge due to the proximity of neurovascular structures and the deep location of PPS tumors, which often interfere with an adequate intra operative visualization and resection of tumors, increasing the risk of complications. For these reasons, over the last 2 decades many Authors focused on the development of minimally invasive or supporting techniques such as endoscopic assistance, surgical robotics and use of surgical navigation systems. Furthermore, a great attention was addressed to the refinement of traditional surgery indications as well as, when indicated, to the adoption of preoperative embolization techniques to obtain possible minimal approaches and reduce mobility.

Transoral approach (Figure 3) has many advantages due to its lower risk of facial nerve impairment, salivary fistulae, no cosmetic drawbacks and shorter hospitalization. However, it is one of the most controversial surgery because of the higher risk of neurologic damage, tumor spillage and severe bleeding.

As for the latter three major aspects, our data support the hypothesis that transoral surgery scenario should be significantly revised, as it could represent a valid treatment for tumors located in lower and prestyloyd portion of PPS. In fact, in our series this approach was associated to lower risk of neurologic complications, no aesthetic impairment, short hospitalization and good surgical outcomes with Low recurrence rate, despite the augmented risk of tumor spillage. Moreover, severe hemorrhage risk was generally low and, when appropriate, preoperative embolization improves outcomes, reducing operative duration and peri operative bleeding rate. In this regard, SQUID12, a novel embolization agent, may be a very promising choice for PPS hyper vascularized tumors, ensuring higher devascularization rate, safer procedures and lower risk of systemic dispersion compared to traditional Contour treatment.

Certainly, there are some limitations to our study. First, this study is based on a monocentric evaluation, thus, the number of cases is relatively small. Furthermore, it must be considered that as a retrospective analysis, our study has some obvious limitations owing to its design. In fact, it depends on a review of charts not originally designed to collect data for research, thus some information is bound to be lacking. Another disadvantage of this type of study is that many different healthcare professionals will have been involved in patient care over a long period of time, so the measurement of many clinical variables would probably be less accurate.

## Figures and Tables

**Figure 1 jpm-13-00283-f001:**
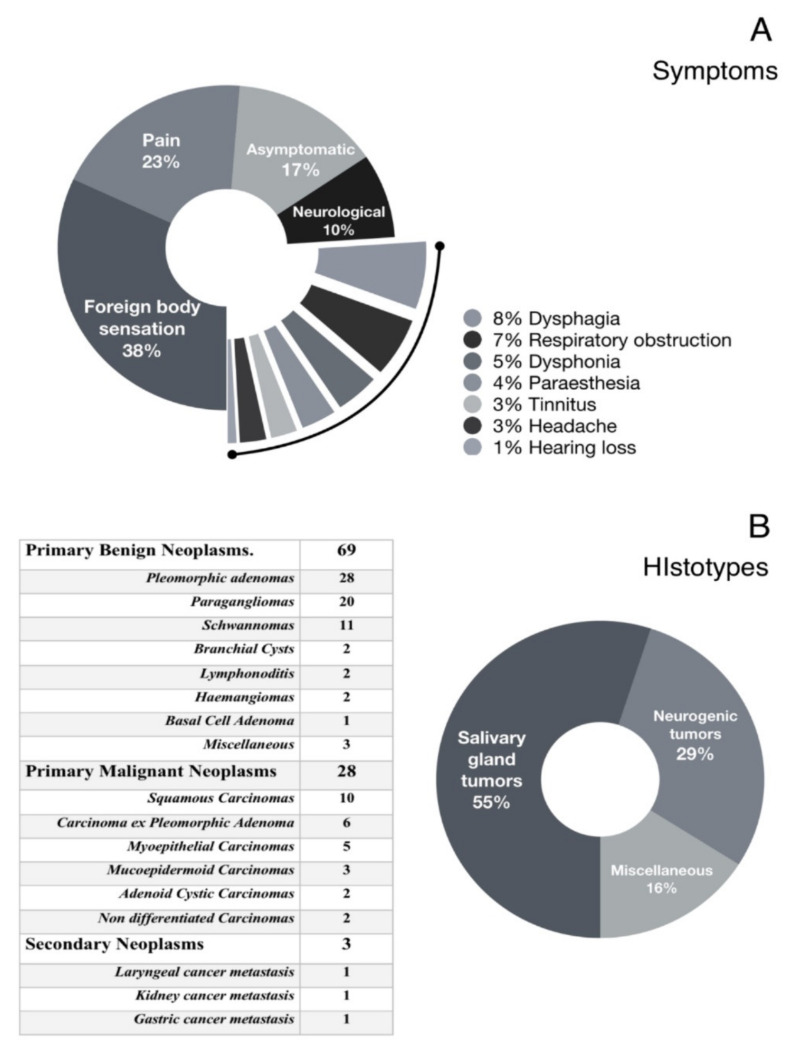
Para-pharyngeal space tumors clinical features. (**A**): main presenting symptoms and clinical signs. (**B**): different tumor types.

**Figure 2 jpm-13-00283-f002:**
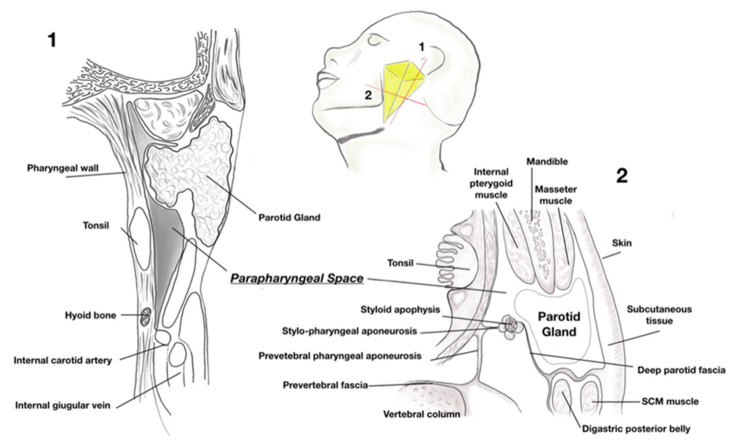
Parapharyngeal space anatomy. Yellow box: representation of pyramid shaped parapharyngeal space. **1**: sagittal section view. **2**: cross section view.

**Figure 3 jpm-13-00283-f003:**
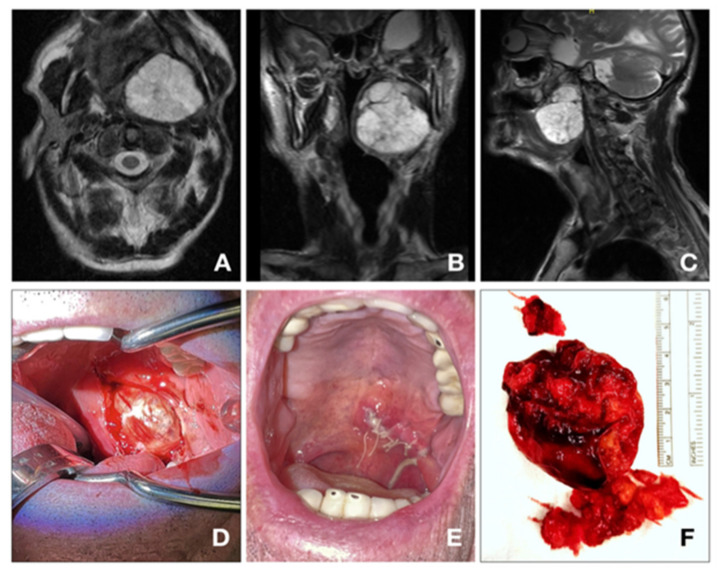
(**A**) 91 y.o. patient, treated for a pleomorphic adenoma of the deep parotid lobe (56 × 49 × 50 mm). (**A**–**C**): MRI t2-TSE ax; (**D**–**F**): Tumor excision/transoral approach).

**Table 1 jpm-13-00283-t001:** Comparison of surgical approaches, tumor size and surgery duration. N: number of cases; MTS (cm): maximum tumor diameter; TM(min): mean surgery duration.

Surgical Approach	*N*	*%*	*MTS* (*cm*)	*TM* (*min*)
**Transmandibular**	15	16	8.3	239.5
**Transcervical**	55	57	3.9	74.6
**Transoral**	26	27	2.9	54.5

**Table 2 jpm-13-00283-t002:** Post-operative complications.

	Transmandibular (N = 15)	Transcervical (N = 55)	Transoral (N = 26)
**Non-neurologic**	**4/15 (26.6%)**	**1/55 (1.8%)**	**3/26 (11.6%)**
Ab-ingestis Pneumonia	**2**	**-**	**-**
Persistent local pain	**1**	**-**	**-**
Dyspnea			
Vomit	**-**	**-**	**1**
Disorientation	**1**	**-**	**-**
Trismus	**-**	**-**	**-**
Hemorrhage	**-**	**-**	**2**
**Neurologic**	**6/15 (40%)**	**10/55 (18.2%)**	** *-* **
Vagus nerve injury	**5**	**-**	**-**
Glossopharyngeal injury	**1**	**-**	**-**
Facial nerve injury	**-**	**6**	**-**
Spinal nerve injury	**-**	**2**	**-**
Recurrent nerve injury	**-**	**2**	**-**
**Aesthetic Outcome**	**5/15 (30%)**	**3/55 (5.5%)**	**-**
Lip numbness	**3**	**-**	**-**
Unsightly scar	**2**	**3**	**-**

## Data Availability

Not applicable.

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
