# Peer review of "Parapharyngeal Space Tumors: Our Experience"

_jpm, 2023, doi:10.3390/jpm13020283_

Round 1
Reviewer 1 Report
Dear author, well done it is a well-designed paper containing more valuable knowledge about para-pharyngeal tumors. However, there is some mixed knowledge in the method section that I signed in the paper PDF file. In addition, there have been some grammatical errors throughout the paper, and I signed some of them in it. Please consider them...

Author Response
Dear Reviewer, we appreciated your revision. You will find our corrections according to your suggestions below.
Point 1: your observational retrospective study is starting from 2002 or 2016? please clear it.
Response 1: we agree original text was a bit confusing. We corrected the text as follow:
"We performed a monocentric, retrospective observational study, on a total of 1800 patients affected by cervical tumors, who underwent to diagnostic and therapeutic workout in our Centre between 2002 and 2021"
Point 2: spelling error, line 156.
Response 2: text revised.
Point 3: spelling error, line 180.
Response 3: text revised.
Point 4: in this sentence, i think there are grammatical faults, please reconsider to it. line 182.
Response 4: we corrected the text as follow: "Parapharyngeal space tumors are rare, accounting for approximatively 0,5/1,5% of all head and neck neoplasms"
Reviewer 2 Report
Dear authors, this is a very interesting retrospective case series of managing parapharyngeal space tumours PPS in your centre. In general, the data is well presented but since this is a single-centre experience article, I would personally expect more information on this. To be precise, at first, the criteria by Riffat should be furtherly analyzed, despite being known to the MDT working on PPS. Furhtermore, I would like to see your input on which of these tumours would you need biopsy prior to surgery, and what are your criteria on that. In addition, what is your algorithm on imaging prior to surgery (CT, MRI, possibly PET-CT on high suspection of malignancy). A short reference to limitation of studies should be appropriate.
Author Response
Dear Reviewer, thank you for your revision, according to which we corrected our manuscript as follows:
Point 1: Dear authors, this is a very interesting retrospective case series of managing parapharyngeal space tumours PPS in your centre. In general, the data is well presented but since this is a single-centre experience article, I would personally expect more information on this. To be precise, at first, the criteria by Riffat should be furtherly analyzed, despite being known to the MDT working on PPS.
Response 1: we added the following text in methods: "We performed a monocentric, retrospective observational study, on a total of 1800 patients affected by cervical tumors, who underwent to diagnostic and therapeutic workout in our Centre between 2002 and 2021. Patients were identified considering a list of International Classification of Disease (ICD) codes that refer to neoplasms involving PPS. According to many inclusion criteria of Riffat et al. (2014), we selected 98 patients, 50 males and 48 females, ranging from 20 months to 86 years of age. More specifically, only tumors located in PPS were considered eligible, while metastatic neoplasms involving PPS but originating from other anatomical regions were excluded. Among the salivary neoplasms of the parotid deep lobe, we included only those located in the retroangulomandibular neck region. Among the carotid paragangliomas, we considered eligible only those located above the posterior belly of the digastric muscle. Cases lacking an accurate description of preoperative diagnostic techniques, adequate report on the tumor characteristics, surgical approach, histological subtypes diagnosis, postoperative outcome or complications accurate report were excluded".
Point 2: Furhtermore, I would like to see your input on which of these tumours would you need biopsy prior to surgery, and what are your criteria on that. In addition, what is your algorithm on imaging prior to surgery (CT, MRI, possibly PET-CT on high suspection of malignancy).
Response 2: we added the following text in 2.1 paragraph: "Although MRI is generally considered the best choice in the evaluation of parapharyngeal tumors as it usually provides better information about soft tissues compared to TC scan, we performed both MRI (76 patients) and CT (51 patients) preoperative examinations in many cases, as they could be complementary in tumor characteristics evaluation, particularly in potentially malignant lesions. MRI was often more able to precisely define the relationship between tumor and surrounding great vessels, especially internal carotid artery (ICA) and to define more accurately the nature of some specific neoplasms (e.g. pleomorphic adenomas, paragangliomas). On the other hand, CT scans provided more accurate information on bony invasion, erosion or calcification within tumors. A selective angiography was preoperatively proposed for all enhancing lesions (37 patients) to favor differential diagnosis between neurogenic and vascular tumors. Ultrasound-assisted fine needle aspiration (USgFNAC) was performed when malignancy was clinically or radiologically suspected (25 patients), with the exception of vascular lesions or in case of inadequate ultrasound targeting or lesion proximity with major vessels. Moreover, in patients who received malignant diagnosis, FDG-PET has been performed as a useful method to detect distant metastasis not only in pre-operative assessment to support treatment planning, but also in mid and late post-operative follow-up. Incisional transoral or transcervical biopsy was rarely considered beneficial in our patients, thus it was performed only in patients who were not candidates for surgery, to obtain the histological diagnosis (3 patients)."
Point 3: A short reference to limitation of studies should be appropriate.
Response 3: we added the followin text in conclusios: "Certainly, there are some limitations to our study. First, this study is based on a monocentric evaluation, thus, the number of cases is relatively small. Furthermore, it must be considered that as a retrospective analysis, our study has some obvious limitations owing to its design. In fact it depends on a review of charts not originally designed to collect data for research, thus some information is bound to be lacking. Another disadvantage of this type of study is that many different healthcare professionals will have been involved in patient care over a long period of time, so the measurement of many clinical variables would probably be less accurate".